# Ti(O*i*-Pr)$_4$-promoted photoenolization Diels–Alder reaction to construct polycyclic rings and its synthetic applications

Baochao Yang[1], Kuaikuai Lin[1], Yingbo Shi[1] & Shuanhu Gao [1]

Stereoselective construction of polycyclic rings with all-carbon quaternary centers, and vicinal all-carbon quaternary stereocenters, remains a significant challenge in organic synthesis. These structures can be found in a wide range of polycyclic natural products and drug molecules. Here we report a Ti(O*i*-Pr)$_4$-promoted photoenolization/Diels–Alder (PEDA) reaction to construct hydroanthracenol and related polycyclic rings bearing all-carbon quaternary centers. This photolysis proceeds under mild conditions and generates a variety of photo-cycloaddition products in good reaction efficiency and stereoselectivity (48 examples), and has been successfully used in the construction of core skeleton of oncocalyxones, tetracycline and pleurotin. It also provides a reliable method for the late-stage modification of natural products bearing enone groups, such as steroids. The total synthesis of oncocalyxone B was successfully achieved using this PEDA approach.

---

[1] Shanghai Key Laboratory of Green Chemistry and Chemical Processes, School of Chemistry and Molecular Engineering, East China Normal University, 3663N Zhongshan Road, Shanghai 200062, China. Correspondence and requests for materials should be addressed to S.G. (email: shgao@chem.ecnu.edu.cn)

olycyclic rings containing anthracyclines, anthracenols and anthraquinones are ubiquitous in natural products and drug molecules. For example, the most well-known members of this family of natural products, doxorubicin (**1**)[1] and tetracycline (**2**)[2–4] (Fig. 1b), have been extensively studied as anticancer and antibiotic drugs. Interestingly, we also noticed a group of polyketides with all-carbon quaternary centers within the core anthracenols or anthraquinones, such as the newly isolated A-74528 (**3**)[5, 6] and JBIR-85 (**4**)[7], as well as the antibiotics pleurotin (**5**)[8–13] and oncocalyxone B (**6**)[14, 15]. The additional quaternary carbons contribute to the diversity of these challenging chemical structures. Not surprisingly, these natural molecules show potential to be new anticancer or antibiotic agents. Notably, pleurotin (**5**) has attracted the attention of chemical and biological communities because of its potent activity against Gram-positive bacteria and its ability to inhibit the cellular redox system based on thioredoxin–thioredoxin reductase (half-maximal inhibitory concentration=170 nM). From the structural point of view, the basic skeletons (I–IV, Fig. 1a) of these natural polycycles feature linearly fused six-membered rings with various oxidation states and consecutive stereogenic centers, which represent the greatest challenge in their chemical synthesis.

To date, the Hauser annulation is one of the most commonly used methods for synthesizing anthraquinones from stabilized phthalides and Michael acceptors (Fig. 2a)[16–19]. This annulation involves tandem Michael addition/Dieckmann cyclization to generate *para*-anthracenols **V**, which can be easily oxidized to anthraquinones **I**. Alternatively, an *ortho*-toluate nucleophile can be used in this tandem process, in which case it is also known as Staunton–Weinreb annulation, and subsequent Michael addition/Dieckmann or Claisen condensation generates hydroanthracenol **VI** in a lower oxidation state (Fig. 2b)[20–22]. Myers and co-workers[23, 24] have used this versatile methodology

to prepare thousands of tetracycline derivatives. However, neither conventional Hauser annulation nor the Staunton–Weinreb variation can be used to construct hydroanthracenol **IV**, which has three consecutive stereogenic centers.

Photo-induced enolization of 2-alkyl benzophenones is an effective method to generate the active diene species hydroxy-*o*-quinodimethane, which can be trapped by electron-deficient dienophiles to form the desired cycloaddition products[25–28]. This methodology was first discovered by Yang and Rivas in 1961[29], and it was further studied and applied in organic synthesis[30–43], especially natural product total synthesis[44–48]. In 2004, Nicolaou and co-workers[44–48] reported further studies of this reaction in both inter- and intra-molecular versions and its elegant applications in the total synthesis of hybocarpone, hamigerans and their derivatives. In their work, mono-substituted and 1,1-disubstituted olefins were used as dienophiles in the intermolecular photoenolization/Diels–Alder (PEDA) reaction, generating the cycloaddition products in moderate stereocontrol. However, using sterically hindered 2,2-disubstituted olefin as a dienophile has been elusive[49–51].

Since our research group is devoted to the synthesis of bioactive natural products, we decided to explore new approaches to prepare the core structure **IV** in a stereo-controlled way, which can serve as a common precursor for the synthesis of others (I–III) with higher oxidation states by simple oxidations. We also hope this method can be effectively utilized in the synthesis of **3**–**6** and their derivatives for biological and drug discovery studies. Here we report a method to synthesize **IV**, in which the target is prepared from 2-methylbenzaldehyde and Michael acceptors via titanium-promoted PEDA reaction. This method allows stereospecific formation of quaternary centers and convergent synthesis of related polycyclic natural products. As far as the intermolecular PEDA reactions are concerned, we realized

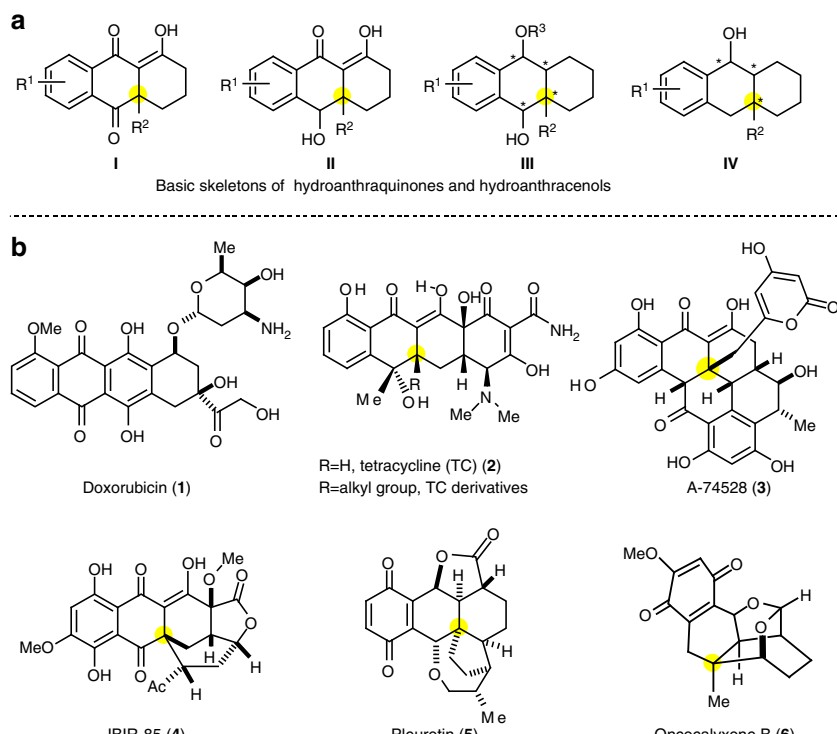

**Fig. 1** Natural products containing anthracenols or anthraquinones. **a** The basic skeletons (**I–IV**) of hydroanthraquinones and hydroanthracenols contain linearly fused six-membered rings with various oxidation states and consecutive stereogenic centers. **b** Doxorubicin (**1**) and tetracycline (**2**) are anticancer and antibiotic drugs containing anthraquinone and anthracenol, respectively. A-74528 (**3**) and JBIR-85 (**4**), pleurotin (**5**) and oncocalyxone B (**6**) are structurally related polyketides bearing all-carbon quaternary centers within the core anthracenols or anthraquinones

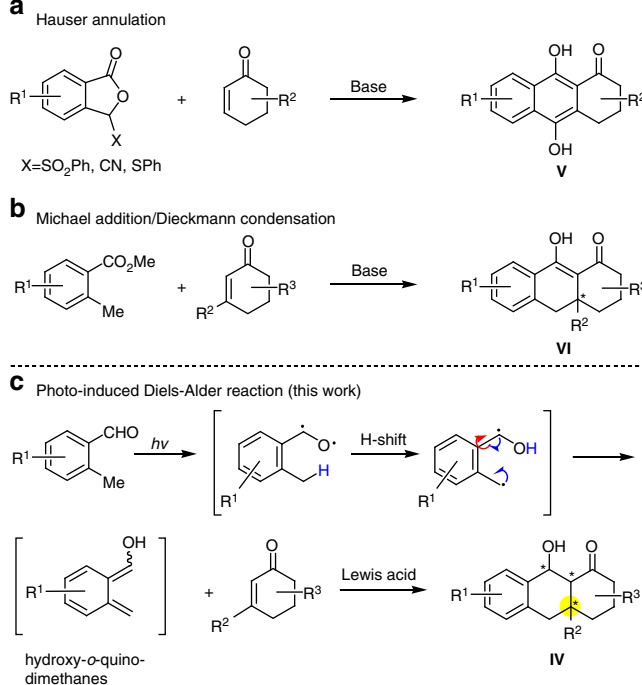

**a** Hauser annulation

X=SO₂Ph, CN, SPh

**b** Michael addition/Dieckmann condensation

**c** Photo-induced Diels-Alder reaction (this work)

hydroxy-*o*-quino-dimethanes

**Fig. 2** Methods for generating core anthracenols and anthraquinones. **a** Hauser annulation is used to prepare *para*-anthracenols **V** through a tandem Michael addition/Dieckmann cyclization process. **b** Staunton–Weinreb annulation is a similar strategy to synthesis hydroanthracenol via a Michael addition/Dieckmann or Claisen condensation. **c** Photo-induced Diels–Alder reaction is an effective method to generate the active diene species hydroxy-*o*-quinodimethane for the intermolecular [4 + 2] cycloaddition

that the major challenges were: how to activate the poorly reactive 2,2-disubstituted dienophiles, and how to stabilize the short-lived photoenolized dienes. We speculated that this problem could be bypassed by adding Lewis or Brønsted acids to induce formation of a bichelated complex involving both diene and dienophile. This would temporarily turn an otherwise intermolecular PEDA into an intramolecular one, which would help to control the diastereoselectivity of the Diels–Alder reaction and stereospecifically form three consecutive stereogenic centers.

## Results

**Optimization of reaction conditions**. We began our PEDA studies (Fig. 2c) using electron-rich 3,6-dimethoxy-2-methyl-benzaldehyde (**7**) and electron-deficient 3-methylcyclohexenone (**8**) as model substrates. When we tested the photoreaction conditions of Nicolaou et al.[48], we found that ultraviolet (UV) irradiation ($\lambda_{max}$=366 nm) of a solution of **7** (1.0 equivalent (equiv.)) and **8** (6.0 equiv.) in degassed toluene for 2 h led to only trace amounts of hetero-cycloaddition product **11** and no formation of the desired cycloaddition product (Table 1, entry 1). This suggested that the active hydroxy-*o*-quinodimethane species of **7** was generated and decomposed soon after photoenolization, before it could interact with the sterically hindered dienophile **8**. We then tried to activate **8** by adding Brønsted or Lewis acids during photolysis. After extensive screening of 27 Brønsted and Lewis acids (entries 2–8, others see Supplementary Table 1), we were pleased to find that stoichiometric amounts of titanium(IV) isopropoxide Ti(O*i*-Pr)₄ (>2.0 equiv.) successfully promoted intermolecular [4 + 2] cycloaddition, giving the desired product **9** as a single diastereomer as well as its dehydration enone **10** in

56% combined yield (entry 8)[52, 53]. The relative configuration of **9** was confirmed by X-ray diffraction analysis. Alternatively, we were also able to substitute Ti(O*i*-Pr)₄ with metal triflate Lewis acid Yb(OTf)₃, but the yield of **9** was very low (entry 7). We further optimized photolysis conditions by extensively screening solvents, additives and light sources in the presence of Ti(O*i*-Pr)₄ (for more details see Supplementary Tables 2–4). UV light at 300 nm gave similar yield as light at $\lambda_{max}$=366 nm, though the relative proportion of dehydrated product was greater (entry 9). Most solvents other than protic ones (e.g., ethanol) worked, including chlorobenzene, dichloromethane, acetonitrile and cyclohexane (entries 12–18). Notably, ether solvents such as diethyl ether and dioxane gave photolytic yields of **9** comparable to that obtained with toluene. Using anhydrous solvents slightly improved yield (entries 11, 17–19). Interestingly, we found that formation of dehydration product **10** was effectively suppressed when freshly recrystallized **7** was used in photolysis, affording **9** in 72% isolated yield. This may be because recrystallization removed small amounts of 3,6-dimethoxy-2-methylbenzoic acid, which can eliminate the $\beta$-hydroxyl ketone of **9**. We confirmed that this PEDA reaction requires both UV light and stoichiometric amounts of the Lewis acid Ti(O*i*-Pr)₄: no reaction occurred when the reaction mixture was stirred without irradiation in the presence of stoichiometric Ti(O*i*-Pr)₄ (entry 20). Obviously, activation of 2-methylbenzaldehyde by UV light initiates subsequent cycloaddition. It has been shown that this intermolecular cycloaddition depends strongly on the stoichiometry of Ti(O*i*-Pr)₄. We found that using a catalytic amount (0.2 equiv.) produced only trace amounts of **9**, while using >2.0 equiv. provided optimal yield.

**Substrate scope**. We then investigated the scope of this PEDA reaction with respect to: the properties of diene (aromatic rings), the diversity of dienophiles and the potential application of the cycloaddition products. Using the $\beta$-substituted cyclohexenones as dienophiles, we planned to examine the generality of this photolysis for the construction of hydroanthracenol and related polycyclic rings (Table 2). We first tested easily available building blocks containing 3-methyl cyclohexenone moieties, such as piperitone (**15**) and isophorone (**16**). Under optimal conditions, photoreaction of **7** with **15** or **16** afforded the corresponding cycloaddition products **24** or **25** as single diastereomers in good yield. These results indicated that the substituents on the cyclohexenone ring did not affect the photolysis. Photolysis also tolerated substrates with hydroxyl and carbonyl functional groups or acid-sensitive protecting groups such as *tert*-butyldimethylsilyl ether (OTBS); in all cases, the desired product **26** with four consecutive stereogenic centers was obtained. Electron density on the diene ring did affect photolysis, with electron-donating substituents facilitating the photoreaction. Irradiation of monomethoxy-, trimethoxy- or fully substituted methylbenzaldehydes (**12–14**) with dieno-philes (**8, 15–17**) stereospecifically generated the desired cycloaddition products (**27–34**) in synthetically useful yields. Notably, reaction efficiency increased with increasing electron density on the aromatic ring. We then explored reactions involving bicyclic dienophiles with two *cis*- or *trans*-fused six-membered rings (**18** and **19**). As we expected, irradiating **7** or **13** with **18** or **19** successfully furnished the desired tetracyclic products (**35–38**) bearing the basic skeleton of tetracycline and an additional quaternary center. The relative stereochemistry of tetracycline **36a** was confirmed by X-ray diffraction analysis. We were pleased to find that photo-cycloaddition of the syn-thetically useful bicyclic building blocks Hajos–Parrish ketone (**20**) or Wieland–Miescher ketone (**21**), both of which contain an

**Table 1 Condition screening of photolysis**

| Entry | $\lambda_{max}$ (nm) | Acid (3.0 equiv.) | Solvent[b] | Time (h) | Conversion (%)[c] | Yield (9 + 10, %)[c] | Yield (11, %)[c] | Ratio (9/10)[c] |
|---|---|---|---|---|---|---|---|---|
| 1 | 366 | None | Toluene | 2.0 | 100 | ND | Trace | – |
| 2 | 366 | TFA | Toluene | 3.5 | 82 | ND | 5.4 | – |
| 3 | 366 | PTSA·H$_2$O | Toluene | 3.5 | 87 | ND | 1.3 | – |
| 4 | 366 | TMSOTf | Toluene | 2.0 | 99 | ND | ND | – |
| 5 | 366 | BF$_3$·Et$_2$O | Toluene | 2.0 | 60 | ND | ND | – |
| 6 | 366 | Sc(OTf)$_3$ | Toluene | 2.0 | 76 | ND | ND | – |
| 7 | 366 | Yb(OTf)$_3$ | Toluene | 2.0 | 98 | 16 | 0.8 | 2.2:1 |
| 8 | 366 | Ti(O$i$-Pr)$_4$ | Toluene | 0.5 | 100 | 56 | 1.6 | 14.3:1 |
| 9 | 300 | Ti(O$i$-Pr)$_4$ | Toluene | 0.75 | 96 | 55 | 3.2 | 4.8:1 |
| 10 | 254 | Ti(O$i$-Pr)$_4$ | Toluene | 1.5 | 98 | 19 | 1.5 | 3.4:1 |
| 11 | 366 | Ti(O$i$-Pr)$_4$ | Toluene (dry) | 0.5 | 100 | 60 | 2.9 | 20:1 |
| 12 | 366 | Ti(O$i$-Pr)$_4$ | Chlorobenzene | 0.5 | 100 | 61 | 3.4 | 12.5:1 |
| 13 | 366 | Ti(O$i$-Pr)$_4$ | CH$_2$Cl$_2$ | 1.5 | 100 | 36 | 5.6 | 1:2.1 |
| 14 | 366 | Ti(O$i$-Pr)$_4$ | CH$_3$CN | 2.0 | 94 | 22 | 8.0 | 6.2:1 |
| 15 | 366 | Ti(O$i$-Pr)$_4$ | EtOH | 1.5 | 30 | ND | Trace | – |
| 16 | 366 | Ti(O$i$-Pr)$_4$ | Cyclohexane | 0.5 | 100 | 67 | 4.8 | 4.8:1 |
| 17 | 366 | Ti(O$i$-Pr)$_4$ | Et$_2$O (dry) | 0.5 | 100 | 61 | 4.0 | 16.7:1 |
| 18 | 366 | Ti(O$i$-Pr)$_4$ | Dioxane (dry) | 0.5 | 100 | 67 | 3.1 | 20:1 |
| 19[d] | 366 | Ti(O$i$-Pr)$_4$ | Dioxane (dry) | 0.5 | 100 | 72[e] | – | 50:1 |
| 20 | None | Ti(O$i$-Pr)$_4$ | Dioxane (dry) | 72.0 | 14 | ND | ND | – |

ND not detected
[a]Conditions: aldehyde (0.1 mmol), unsaturated ketone (0.6 mmol), conc.=0.02 mol/L
[b]All the photoreactions were conducted in degassed solvent
[c]Conversions, ratios and yields were determined by $^1$H NMR analysis of crude mixtures using CH$_2$Br$_2$ as an internal standard, unless noted
[d]Recrystallized aldehyde (0.5 mmol) was used
[e]Isolated yield

enone group at the ring junction, smoothly gave rise to the desired tetracyclic rings (**39–42**) in moderate yields (24–61%) under optimal conditions. The chemical structure of compound **39** and its relative configuration were determined by X-ray diffraction analysis: this compound has the core structure and stereochemistry of pleurotin (**5**)[5], potentially making it a useful intermediate en route to this challenging molecule and its derivatives. Moreover, two steroids, 4-cholesten-3-one (**22**) and norandrostenedione (**23**), were selected as dienophiles in this PEDA reaction. We were surprised to find that the photoreaction worked well and formed the expected hexacyclic rings (**43–46**) in 59–76% yield. The structures of **44** and **45b** were confirmed by X-ray diffraction analysis. These results highlight the ability of this PEDA approach to generate synthetically challenging steroid-containing architectures, which could potentially serve as steroid derivatives or lead compounds for biological or drug discovery studies. It also shows the PEDA reaction to be an efficient method for late-stage modification of enone-containing natural products and drug molecules.

During our studies of reaction scope, we found that 2-methylbenzaldehyde substrates lacking a methoxy group *ortho* to the aldehyde showed no reactivity in this PEDA reaction. This identifies the *ortho*-methoxy group as crucial for the reaction, consistent with observations by Nicolaou et al.[48]. The *ortho*-methoxy group was not required in the case of less sterically hindered dienophiles, as Nicolaou et al.[48] also found. We considered that the photoenolized hydroxy-*o*-quinodimethane and *ortho*-methoxy group may be chelated by Ti (O$i$-Pr)$_4$, forming a relatively stable complex. The *ortho*-methoxy may serve as a key neighboring group that helps to stabilize the short-lived photoenolized hydroxy-*o*-quinodimethane diene. Meanwhile, according to the findings of condition screening, using >2.0 equiv. Ti(O$i$-Pr)$_4$ provided optimal yield. These results suggest, taking model substrate as an example, that the thereaction is driven by complexation of Ti(O$i$-Pr)$_4$ with both reactants **7** and **8**. Next, we planned to further explore the generality and stereoselectivity of this PEDA reaction using 1,1-disubstituted olefins as dienophiles (Table 3). Both cyclic and acyclic olefins bearing a variety of electron-withdrawing groups, such as ketones (**47–50**), aldehydes (**51–53**) and esters (**54** and **55**), were investigated through the photo-cycloaddition with substituted 2-methylbenzaldehyde (**7**, **12–14**). We found that a mixture of **7** and 2-methyl cyclohexenone (**47**) or (**48**) underwent the Ti-promoted PEDA process and smoothly produced the desired products in good yields with *endo/exo* stereoselectivity ranging from 3:1 to 7:1. The photoreaction between **7** and **49** or **50** under the same conditions yielded **58** or **59**, respectively, as single diastereomers. The structure and relative stereochemistry of **56a** and **59** were confirmed by X-ray diffraction analysis. Meanwhile, as a control experiment, we also systematically investigated the reaction efficiency of **7** and **49**, and the yield of the reaction increased obviously when the amount of Ti(O$i$-Pr)$_4$ increased from catalytic to stoichiometric (for more details see Supplementary Table 7). These results

**Table 2 Scope of the PEDA reaction involving β-substituted cyclohexenones as dienophiles**

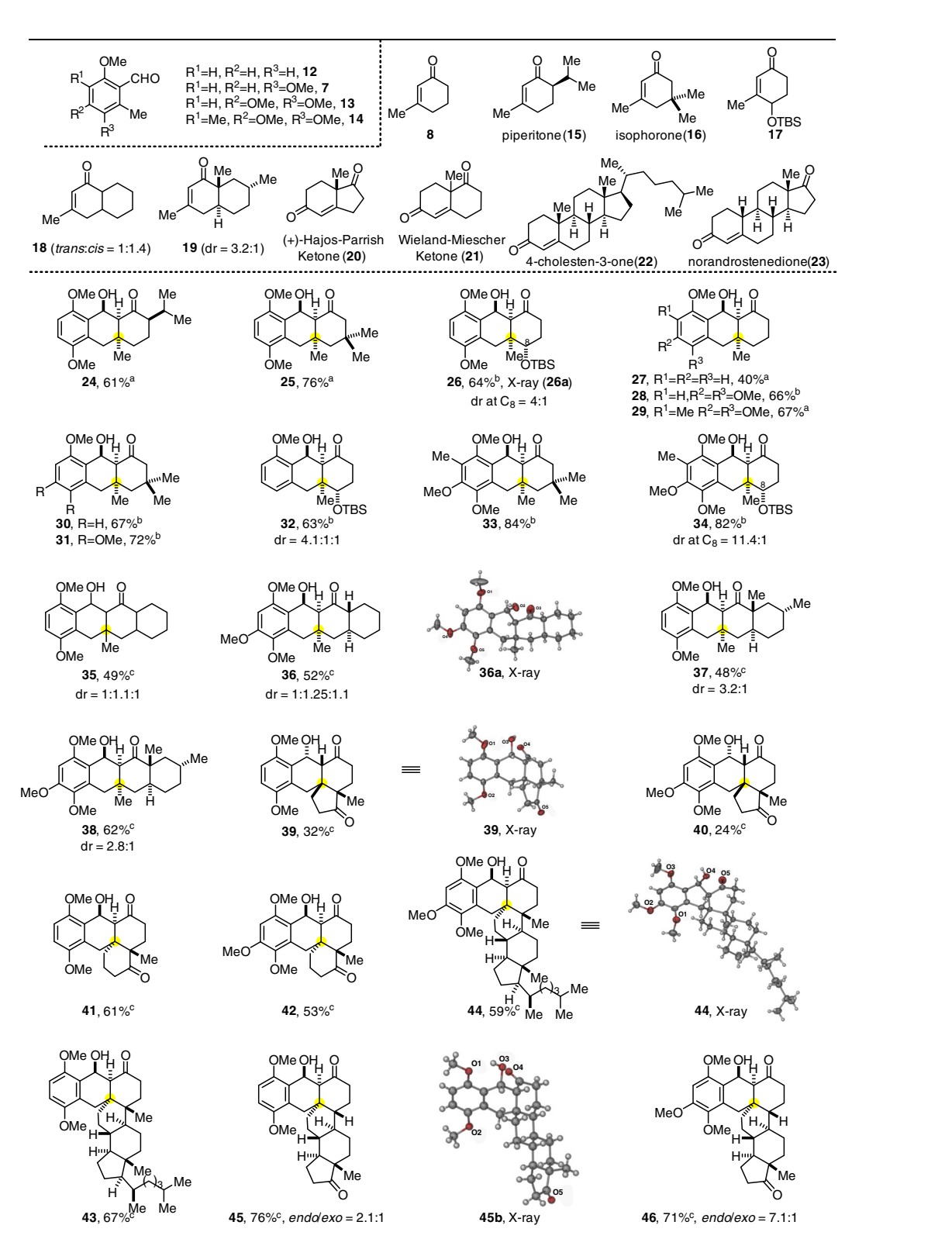

aMethod A: hv (λmax=366 nm), Ti(Oi-Pr)₄ (3.0 equiv.), diene (0.3 mmol, 1.0 equiv.), dienophile (6.0 equiv.), conc.=0.02 mol/L, dioxane (anhydrous and degassed)
bMethod B: hv (λmax=366 nm), Ti(Oi-Pr)₄ (1.2 equiv.), diene (0.3 mmol, 1.0 equiv.), dienophile (1.5 equiv.), conc.=0.02 mol/L, toluene (anhydrous and degassed)
cMethod C: hv (λmax=366 nm), Ti(Oi-Pr)₄ (3.0 equiv.), dienophile (0.3 mmol, 1.0 equiv.), diene (2.0–4.0 equiv.), toluene (anhydrous and degassed)

**Table 3 Scope of the PEDA reaction involving 1,1-disubstituted olefins as dienophiles**

**56**, 67%[a]
*endo/exo* = 3:1

**56a**, X-ray

**57**, 83%[a]
*endo/exo* = 7:1

**58**, 79%[a]

**59**, 78%[a] ≡ **59**, X-ray

**60**, 75%[a]
*endo/exo* = 3:1

**61**, R=H, 82%[a]
*endo/exo* = 1.7:1
**62**, R=Me, 80%[a]
dr = 6.4:1:4.2:5

**63**, 59%[b]

**64**, 78%[a] ≡ **64**, X-ray

**65**, R=Ac, 71%[c]
**66**, R=CHO, 84%[a]
*endo/exo* = 5.3:1

**67**, 79%[b]

**68**, R=Ac, 72%[a]
**69**, R=CHO, 89%[c]
*endo/exo* = 9.3:1

**71**, R=Ac, 74%[c]
**72**, R=CHO, 87%[c]
*endo/exo* = 7.2:1

**70**, R = H, 81%[b]
**73**, R = Me, 65%[b]

[a]Method A: *hv* (λ$_{max}$=366 nm), Ti(O*i*-Pr)$_4$ (3.0 equiv.), diene (0.3 mmol, 1.0 equiv.), dienophile (6.0 equiv.), conc.=0.02 mol/L, dioxane (anhydrous and degassed)
[b]Method B: *hv* (λ$_{max}$=366 nm), Ti(O*i*-Pr)$_4$ (3.0 equiv.), diene (0.3 mmol, 1.0 equiv.), dienophile (6.0 equiv.), conc.=0.02 mol/L, toluene (anhydrous and degassed)
[c]Method C: *hv* (λ$_{max}$=366 nm), Ti(O*i*-Pr)$_4$ (1.2 equiv.), diene (0.3 mmol, 1.0 equiv.), dienophile (1.5 equiv.), conc.=0.02 mol/L, toluene (anhydrous and degassed)

indicate that the 1,1-disubstituted olefins used here, especially cyclic ones, also require the activation of Lewis acid Ti(O*i*-Pr)$_4$, which were in analogy to β-substituted cyclohexenones. Then, we turned our attention to aldehydes: all unsaturated aldehydes that we tested reacted well, giving *endo/exo* mixtures of the corresponding products **60–62**. When the unsaturated ester **55** or lactone **54** were employed, the respective cycloaddition products **63** or **64** were acquired stereospecifically in good yield. Combining the dienophiles **49**, **51** or **54** with different photoenolized dienes led to the expected products (**65–73**) with similar reaction efficiency and stereoselectivity. These results suggest that the structure of dienophiles as well as the properties of electron-withdrawing functional groups dramatically affect the stereoselectivity of photocycloaddition.

Additionally, we also tested 1,1,2,2-tetra-substituted olefins as dienophiles in this PEDA reaction in order to form products with contiguous all-carbon quaternary centers (Table 4). Construction of vicinal all-carbon quaternary stereocenters is extremely challenging in organic synthesis and natural product total synthesis[54–56] because of the severe steric constraints. We first selected 2,3-dimethylcyclohexenone (**74**), which has one more methyl group than the model dienophile **8**, in order to investigate its reactivity and stereoselectivity of the PEDA reaction. We were pleased to find that photolysis of **7** or **13** with **74** in the presence of Ti(O*i*-Pr)$_4$ under optimal conditions gave the desired products **78** and **79** as single diastereomers in respective yields of 38 and 56%. These products have the same relative stereochemistry as **9**, based on single-crystal diffraction analysis of **78**. We then

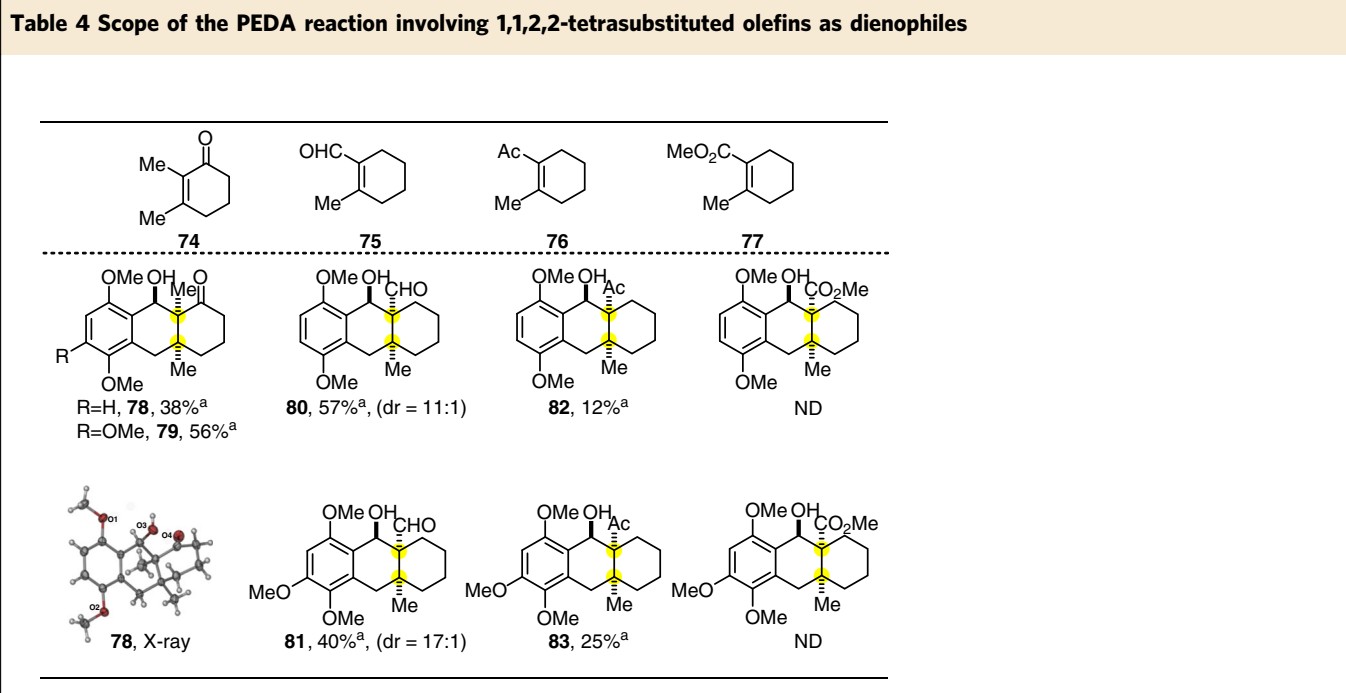

**Table 4 Scope of the PEDA reaction involving 1,1,2,2-tetrasubstituted olefins as dienophiles**

ND not detected
[a]Method D: *hv* ($\lambda_{max}$=366 nm), Ti(O*i*-Pr)$_4$ (3.0 equiv.), dienophile (0.4 mmol, 1.0 equiv.), diene (4.0 equiv.), dioxane (anhydrous and degassed)

prepared three comparable substrates **75–77** with aldehyde, ketone or ester groups in order to study their influence on substrate reactivity in the PEDA process. We found that intermolecular photoreaction between **7** or **13** and dienophiles **75** or **76** underwent Ti-promoted PEDA reaction, producing the tricyclic products **80–83** in low to moderate yields. No desired cycloaddition products were observed under the same conditions when the photoreaction was performed between **7** or **13** and substrate **77**, which contains an unsaturated ester group. These results indicate that the photoreaction of sterically hindered substrates depends strongly on the properties of electron-withdrawing groups on the dienophiles. Reaction efficiency increased with increasing electro negativity of the corresponding functional groups. Notably, this method generates products containing vicinal all-carbon quaternary stereocenters at the ring junctions (Table 4). Such molecules are not easily accessible through Hauser reactions, Staunton–Weinreb annulations or classical Diels–Alder reactions.

**Total synthesis of oncocalyxone B**. We believe that this PEDA reaction provides a solid basis for the preparation of hydroanthracenol and related polycyclic rings bearing all-carbon quaternary centers. To demonstrate the synthetic utility of this approach, we applied it to the total synthesis of oncocalyxone B, a cordiachrome family of unusual meroterpenoids isolated from the wood of *Auxemma oncocalyx* by Carvalho, Braz-Filho and co-workers in 1993 (Fig. 3a)[14, 15]. The bark of this plant is used as indigenous medicine to treat cuts and wounds. Biogenetically related molecules include oncocalyxone A (**84**)[14], B (**6**)[14], C (**85**)[15], D (**86**)[15] as well as glaziovianol (**87**)[57, 58] (Fig. 3a). Oncocalyxone B (**6**) has a more challenging structure than the other members of this family with a hydroanthracenone skeleton and five consecutive stereogenic centers, including one all-carbon quaternary center at C-8a. An acetal moiety between two hydroxyl groups at C-8 and C-10 strains the ring skeleton.

Our synthesis started from the construction of the basic tricyclic ring via intermolecular PEDA reaction of 3,4,6-trimethoxy-2-methylbenzaldehyde (**13**) and electron-deficient 4-(*tert*-butyldimethylsilyloxy)-3-methylcyclohex-2-enone (**17**) under optimal conditions. UV irradiation ($\lambda_{max}$=366 nm) of a solution of **13** and **17** in the presence of Ti(O*i*-Pr)$_4$ in degassed anhydrous toluene gave the desired product **88** in 76% yield (Fig. 3b). The relative stereochemistry of the four stereocenters was determined by X-ray diffraction analysis, which showed the orientation of the hydroxyl group at C-8 to be opposite to that in the naturally occurring molecule. To invert the C-8 stereocenter, the benzylic alcohol at C-10 was protected as the corresponding methoxymethyl ether, and the carbonyl group at C-5 was converted to a methylene group using Tebbe olefination, affording tricyclic compound **90** in 56% yield over two steps. Selective deprotection of the *tert*-butyldimethylsilyl ether followed by Dess–Martin oxidation and substrate-controlled reduction produced **91** bearing the desired hydroxyl group with good diastereoselectivity. During these efforts, we found **91** to be unstable in dichloromethane: it spontaneously transformed into **92**, which has an oxygen bridge. The structure of **92** was unambiguously confirmed by X-ray diffraction analysis. We attributed the formation of **92** to the generation of a benzylic cation that underwent intramolecular trapping by the hydroxyl group at C-8. The methylene group at C-5 in **92** was functionalized into an α-aldehyde group, then we easily inverted the stereochemistry to the desired β-configuration (**94**) under basic conditions. Next, we screened various Lewis or Brønsted acids to install the bridged acetal group via a ring opening/acetalation process. We found that SnCl$_4$[59, 60] smoothly catalyzed this tandem process and led to the formation of **95** in 85% yield. Subsequent CAN oxidation afforded oncocalyxone B (**6**) in 92% yield; [1]H and [13]C nuclear magnetic resonance (NMR) spectra and high-resolution mass spectrometry data for this synthetic compound were fully consistent with those of the natural product[14, 15].

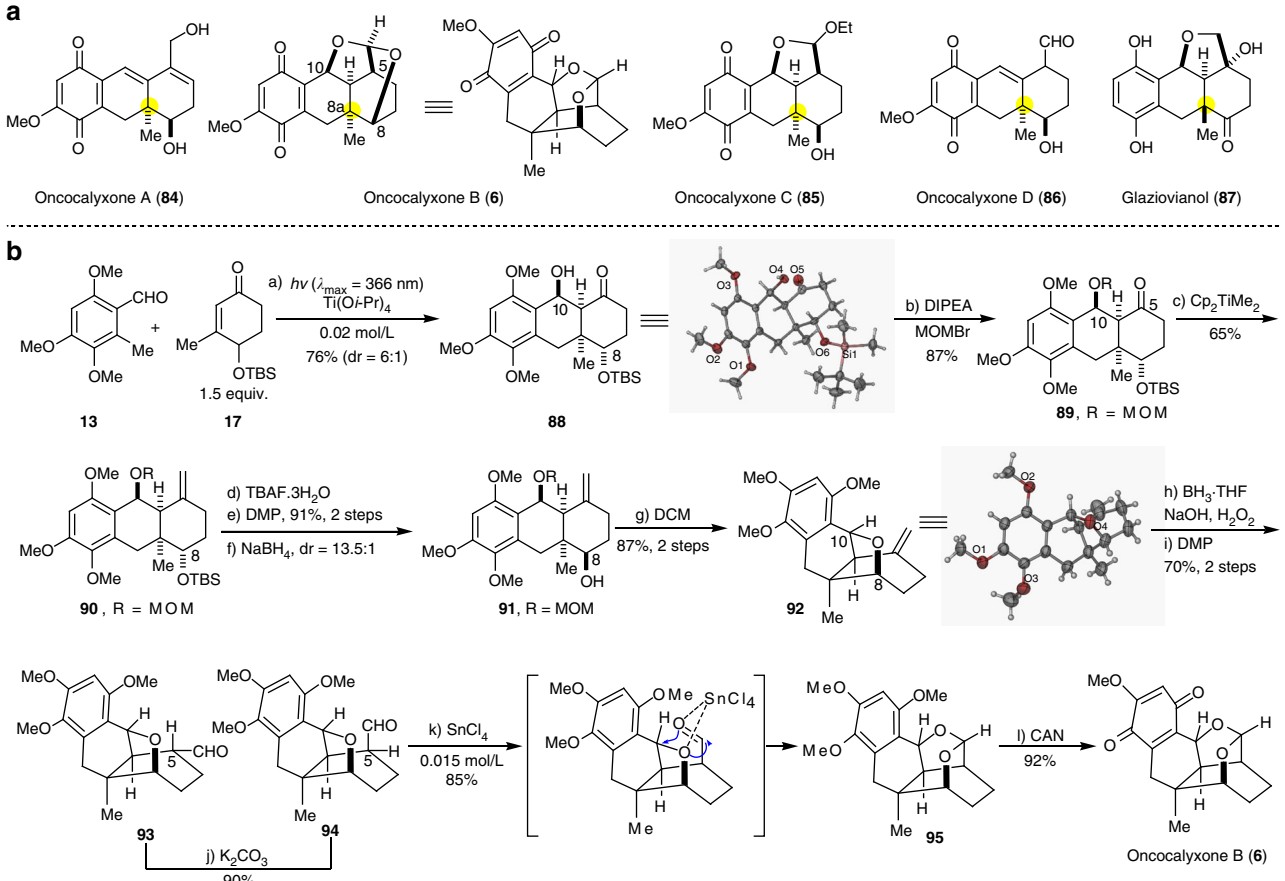

**Fig. 3** Synthetic application. **a** Structurally related natural products contain hydroanthracenol and related polycyclic rings and all-carbon quaternary centers. **b** Total synthesis of oncocalyxone B. (a) $hv$ ($\lambda_{max}$=366 nm), Ti(O$i$-Pr)$_4$, toluene, dr = 6:1, 76%; (b) DIPEA, MOMBr, DCE, 80 °C, 87%; (c) Cp$_2$TiMe$_2$, toluene, 80 °C, 65%; (d) TBAF·3H$_2$O, DMF, 90 °C; (e) DMP, DCM, RT, 91% over 2 steps; (f) NaBH$_4$, MeOH, 0 °C, dr = 13.5:1; (g) DCM, RT, 87% over 2 steps; (h) BH$_3$·THF, 3N NaOH, H$_2$O$_2$, 0 °C; (i) DMP, DCM, RT, 70% over 2 steps; (j) K$_2$CO$_3$, EtOH (degassed), RT, 90%; (k) SnCl$_4$, DCM (0.015 M), −78 °C to 0 °C, 85%; (l) CAN, CH$_3$CN/H$_2$O (2:1), 0 °C, 92%. DIPEA: diisopropylethylamine; MOMBr: bromomethyl methyl ether; DMP: Dess–Martin periodinane; CAN: ceric ammonium nitrate

## Discussion

In summary, a Ti(O$i$-Pr)$_4$-promoted PEDA reaction was systematically studied. We found that stoichiometric amounts of the Lewis acid [Ti(O$i$-Pr)$_4$] plays a key role in this photolysis. The advances of this developed PEDA reaction over previously reported methods rely on the high stereoselectivity and broad reaction scope. UV irradiation ($\lambda_{max}$=366 nm) of mixtures of electron-rich 2-methylbenzaldehyde substrates bearing an *ortho*-methoxy group and various electron-deficient 1,1- or 2,2-disubstituted olefins in the presence of Ti(O$i$-Pr)$_4$ smoothly generated hydroanthracenol and related polycyclic rings with good reaction efficiency and stereoselectivity. To the best of our knowledge, this is the first report that 2,2-substituted and 1,1,2,2-tetra-substituted olefins can serve as dienophiles in a PEDA reaction. This photoreaction is particularly useful for preparing hydroanthracenol and related structures with all-carbon quaternary centers, and even with vicinal all-carbon quaternary stereocenters, which has been demonstrated in the construction of core skeleton of tetracycline and pleurotin. This photoreaction also provides a reliable method for the late-stage modification of enone-bearing natural products such as steroids. This PEDA method was successfully used as a key reaction in the total synthesis of oncocalyxone B. We are currently studying the asymmetric PEDA reaction and its further applications to the total synthesis of complex natural products such as A-74528, JBIR-85 and pleurotin.

## Methods

**General.** For $^1$H and $^{13}$C NMR spectra of the compounds in this article, see Supplementary Figs 1–82, and more experimental details and compound characterization data can be found in Supplementary Methods.

**General procedure for the titanium(IV)-promoted PEDA reaction.** To a solution of aromatic aldehyde (0.3 mmol, 1.0 equiv.) in anhydrous and degassed 1,4-dioxane (15 mL, 0.02 M) in quartz tube sealed with rubber plug was added dienophile (1.8 mmol, 6.0 equiv.) (if the dienophile was solid, it was added before the solvent) under N$_2$, then titanium(IV) isopropoxide (0.9 mmol, 3.0 equiv.) was added, and after homogeneous mixing, the solution was photolyzed at room temperature in a Rayonet chamber reactor (16 lamps) at $\lambda_{max}$=366 nm for certain time. Then, the reaction mixture was poured into saturated sodium bicarbonate and stirred over 30 min, the above mixture was extracted three times with ethyl acetate and the combined organic phases were washed twice with brine and dried over anhydrous sodium sulfate. The dried solution was filtered and the filtrate was concentrated under vacuum. The residue was purified by silica gel column chromatography to give the corresponding product. We generally used 6.0 equiv. dienophiles in the PEDA reactions because they are easily available and have small molecular weight. Such reactions were easy to handle. In reality, 1.5 equiv. dienophiles is sufficient for the reaction. For additional procedures see Supplementary Methods.

**Data availability**. The X-ray crystallographic coordinates for structures reported in this article have been deposited at the Cambridge Crystallographic Data Centre (CCDC). The data can be obtained free of charge from The Cambridge Crystallographic Data Centre via http://www.ccdc.cam.ac.uk/data_request/cif. The authors declare that the data supporting the findings of this study are available within the article. All other data are available from the authors upon reasonable request.

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

## Acknowledgements

We thank the National Basic Research Program of China (973 Program 2015CB856600), National Natural Science Foundation of China (21422203) and National Young Top-Notch Talent Support Program for generous financial support.

## Author contributions

S.G. and B.Y. conceived the project and the synthetic route. S.G. directed the project. B.Y., K.L. and Y.S. conducted the work. S.G., B.Y. and K.L. analysed the experimental results. S.G. wrote the manuscript.

## Additional information

**Competing interests:** The authors declare no competing financial interests.

