## [Peer Review File · Nature Communications]

Reviewers' comments:

Reviewer #1 (Remarks to the Author):

This manuscript describes a titanium-promoted photoenolization/Diels-Alder (PEDA) reaction that constructs polycyclic rings with all-carbon quaternary stereocenters. Polycyclic rings are found in a wide range of complex natural products and pharmaceuticals. In addition, the construction of quaternary stereocenters remains a difficult problem in organic synthesis. Therefore, it is of much value to develop a method for stereoselective formation of polycyclic rings bearing quaternary stereocenters. The authors found that titanium(IV) species plays a key role in the PEDA reaction to construct polycyclic rings. However, this idea has been already reported by Nicolaou and co-workers (ref 37). The difference is that di- or tetrasubstituted olefins can be used as dienophiles. So, there is neither special chemical innovation nor unique originality in this work. As a consequence, although this work is a nice piece of chemical work, this referee has difficult time viewing the work overall as being of the high novelty necessary in Nature Communications; nonetheless, publication in a more specialized journal should be envisioned. This referee admires extensive experiments in this work.

Reviewer #2 (Remarks to the Author):

This is an impressive body of work. While the overall process for Diels-Alder reactions of photoenols is not novel and have been utilized in a number of syntheses, the authors were able to implement this synthetic methodology (and optimize the Lewis acid catalyst) for rapid access to privileged structures. It will definitely be of interest to the broader Nat. Comm. readership, so publication is recommended.

I'll focus my review on a few aspects that need revision/attention, to improve the paper. The mechanism of the [4+2] cycloadditions of "photoenols" generated from aromatic o-methyl ketones is believed to be a "normal electron demand" Diels-Alder, where the electron-rich photoenol is acting as a diene, while the dienophile is an electron-depleted alkene.

Page 3 "(2) how to stabilize the short-lived photoenolized dienes." (->photoenolized)

The role of Ti(OR)₄ as a stabilizer for the photoenol intermediate is questionable. In the introduction the authors quoted Kutateladze's JACS 2016 paper as an example of photoenolization. However, these are aza-counterparts of Nicolaou's all-carbon xylylenes. The aza-counterparts of photoenols undergo cycloadditions which are mechanistically reminiscent of the INVERSE electron demand Diels-Alder reactions. In Angew Chem 2015 11516 paper Kutateladze also used Ti(OR)₄, but this was to enhance the acceptor properties of the azaxylylene and accelerate their reactions with the tethered dienophiles that are donors (because this is an inverse electron demand cycloaddition).

Contrary to this, in the Gao's manuscript, the all-carbon hydroxy-o-quinodimethanes shown in Scheme 1 (eq3) are donors, reacting with unsaturated ketones in a conventional Diels-Alder fashion (unless the authors suggest that the cycloaddition is actually photochemical, i.e. happens in the excited singlet or triplet manifold). The Lewis acid catalyst is needed to activate the dienophile - cyclohexenone. And this part of the rationale is OK. However, is there any specific evidence that Ti(OR)₄ exerts any stabilizing effect on the photoenol? -- besides the >2:1 stoichiometry of Ti(OR)₄ which, in the authors' words, "suggests that the reaction is driven by complexation of Ti(Oi-Pr)₄ with both reactants 7 and 8." The complexation is, I suppose, important to bring to reagents in the immediate proximity of each other (implementing a pseudo-unimolecular reaction), but there is no evidence that the lifetimes of the enols are extended.

The "Ti-promoted photoenolization Diels-Alder" (in red in the TOC graphics) could be interpreted in more than one way - the statement needs to be more concise.

On page 3 the authors describe a photochemical experiment in the absence of a Lewis acid: "This suggested that the active hydroxy-quinodimethane species of 7 was generated and consumed soon after photoenolization." The word "consumed" is somewhat imprecise. Is there a side product, or the photogenerated quinodimethane simply reverts back to the starting ketone via the ground state proton transfer?

Minor points:

The excessive precision for the wavelength (366nm) of irradiation should be corrected. If the authors used Rayonet RPR-3500 lamps, they emit in a broad range from 300 to 400nm, with the max emission somewhere in the middle. The precise label "366nm" is only applicable to narrow bandpass filters, or monocromators, or +/- 5nm UV-LEDs.

There was a minor misassignment of carbons in the Phytochemistry 1995 paper on oncocalyxone B: two pairs need a swap: C4a-C9a and C6-C7.

C4a should have chemical shift of 137.3ppm and the adjacent C9a - 143.1ppm

C6: 15.5ppm; C7: 20.4ppm

To double-check the assignment, the authors could use any method for computing ¹³C shifts listed on Dean Tantillo's webpage cheshirenmr.info

Alternatively, a simple HMBC would show that the carbon at 137.3ppm has an HMBC cross peak with H3 (which is only possible for C4a, not C9a)

"protonic" solvent -> protic

Reviewer #3 (Remarks to the Author):

The authors provide a creative and innovative method for the synthesis of very important class of bioactive natural products that includes tetracyclines and other anthraquinones. It is uniquely suitable for the synthesis of the members containing quaternary stereocenters. The method is based on Lewis acid-mediated photoenolization/Diels-Alder cyclization, and it has been applied in the total synthesis of oncocalyxone B detailed in this manuscript. Although this work is somewhat aligned with the precedent by Nicolaou and co-workers, (J. Am. Chem. Soc. 2004, 126, 613), who reported similar studies down to the use of Ti additives and formation of quaternary stereocenters, I find this work to be of high quality, thorough, and meritorious, strengthened by the total synthesis application. Therefore, I recommend publication after minor editions noted below.

p1. Change sentence "The additional quaternary carbons contribute and demonstrate the diversity of these challenging chemical structures" to "The additional quaternary carbons contribute to the diversity of these challenging chemical structures."

p.3 and throughout the manuscript, replace "stereo-hindered" with "hindered" or "sterically hindered"

p3: replace "we realized that the major challenges rely on:" with "we realized that the major challenges were:"

p3: replace "low-reactice" with unreactive or poorly reactive.

p4. "We found that using a catalytic amount (0.2 equiv.) produced only trace amounts of 9, while using >2.0 equiv. provided optimal yield. These results suggest that the reaction is driven by complexation of Ti(Oi-Pr)₄ with both reactants 7 and 8." It is not clear why the authors conclude that complexation with BOTH is occurring, simple based on stoichiometry of the Ti reagent, as there could be many other explanations. The only solid conclusion is that the Ti reagent is

required, and there is a potential complexation with at least one cycloaddition counterpart. In the absence of more rigorous physical evidence, the sentence needs to be changed to reflect observations more precisely.

Response to Reviewer 1's Comments

1. This manuscript describes a titanium-promoted photoenolization/Diels-Alder (PEDA) reaction that constructs polycyclic rings with all-carbon quaternary stereocenters. Polycyclic rings are found in a wide range of complex natural products and pharmaceuticals. In addition, the construction of quaternary stereocenters remains a difficult problem in organic synthesis. Therefore, it is of much value to develop a method for stereoselective formation of polycyclic rings bearing quaternary stereocenters. The authors found that titanium(IV) species plays a key role in the PEDA reaction to construct polycyclic rings. However, this idea has been already reported by Nicolaou and co-workers (ref 37). The difference is that di- or tetrasubstituted olefins can be used as dienophiles. So, there is neither special chemical innovation nor unique originality in this work. As a consequence, although this work is a nice piece of chemical work, this referee has difficult time viewing the work overall as being of the high novelty necessary in Nature Communications; nonetheless, publication in a more specialized journal should be envisioned. This referee admires extensive experiments in this work.

Although the photoenolized Diels-Alder reactions (PEDA) has been extensively studied and applied since its first discovery by Yang and Rivas in 1961. However, the major challenge of this useful reaction was unsolved, that is the sterically hindered dienophiles cannot be used in the conventional photoenolized Diels-Alder reactions. This drawback extensively restricts its application in the synthesis complex molecules and natural products. In this manuscript, we innovatively introduced Lewis acid to the reaction, which not only activates sterically hindered dienophiles and weakly reactive dienophiles, but also helps to stabilize the photo-enolized hydroxy-o-quinodimethane species. The advances of this newly developed PEDA reaction over previously reported methods, rely on the high stereoselectivity and broad reaction scope. To the best of our knowledge, this is the first report that 2,2-substituted and 1,1,2,2-tetrasubstituted olefins can serve as dienophiles in a PEDA reaction. This photoreaction is particularly useful for preparing hydroanthracenol and related structures with all-carbon quaternary centers, and even with vicinal all-carbon quaternary stereocenters. This photoreaction also provides a reliable

method for the late-stage modification of enone-bearing natural products such as steroids. We believe this breakthrough will open the door for related studies and synthetic applications.

Response to Reviewer 2's Comments

1. This is an impressive body of work. While the overall process for Diels-Alder reactions of photoenols is not novel and have been utilized in a number of syntheses, the authors were able to implement this synthetic methodology (and optimize the Lewis acid catalyst) for rapid access to privileged structures. It will definitely be of interest to the broader Nat. Comm. readership, so publication is recommended.

We thank reviewer 2 for her/his kind recommendation.

2. Page 3 “(2) how to stabilize the short-lived photoenolized dienes.” (→photoenolized)

It was corrected as suggested.

3. The mechanism of the [4+2] cycloadditions of “photoenols” generated from aromatic *o*-methyl ketones is believed to be a “normal electron demand” Diels-Alder, where the electron-rich photoenol is acting as a diene, while the dienophile is an electron-depleted alkene. The role of $\text{Ti}(\text{OR})_4$ as a stabilizer for the photoenol intermediate is questionable. In the introduction the authors quoted Kutateladze's JACS 2016 paper as an example of photoenolization. However, these are aza-counterparts of Nicolaou's all-carbon xylylenes. The aza-counterparts of photoenols undergo cycloadditions which are mechanistically reminiscent of the INVERSE electron demand Diels-Alder reactions. In Angew Chem 2015 11516 paper Kutateladze also used $\text{Ti}(\text{OR})_4$, but this was to enhance the acceptor properties of the azaxylylene and accelerate their reactions with the tethered dienophiles that are donors (because this is an inverse electron demand cycloaddition) Contrary to this, in the Gao's manuscript, the all-carbon hydroxy-*o*-quinodimethanes shown in Scheme 1 (eq3) are donors, reacting with unsaturated ketones in a conventional Diels-Alder fashion (unless the authors suggest that the cycloaddition is actually photochemical, i.e. happens in the excited singlet or triplet manifold). The Lewis acid catalyst is needed to activate the dienophile - cyclohexenone. And this part of the rationale is OK. However, is there any specific evidence that $\text{Ti}(\text{OR})_4$ exerts any stabilizing effect on the photoenol? -- besides the >2:1 stoichiometry of $\text{Ti}(\text{OR})_4$ which, in the authors' words, “suggests that the reaction is driven by complexation of $\text{Ti}(\text{O}i\text{-Pr})_4$ with both reactants 7 and 8.” The complexation is, I suppose, important to bring to reagents in the immediate proximity of each other (implementing a pseudo-unimolecular reaction), but there is no evidence that the lifetimes of the enols are extended.

(1) We agree with this reviewer that the cycloaddition with photo-enolized hydroxy-*o*-quinodimethane in our manuscript is indeed a “normal electron demand” Diels-Alder reaction, and the cycloaddition with photo-generated azaxylylenes or amino azaxylylenes, reported by Kutateladze group, is an inverse electron demand Diels-Alder reaction. $\text{Ti}(\text{O}i\text{-Pr})_4$ plays key roles in both cases. $\text{Ti}(\text{O}i\text{-Pr})_4$ helps to further polarize and activate the azaxylylenes, which dramatically accelerated the cycloaddition. More related papers have been cited (ref. 36-39) to help the readers understanding the progress of this field.

(2) In our case, we believe that $\text{Ti}(\text{O}i\text{-Pr})_4$ not only activates weakly reactive dienophiles, but also helps to stabilize the photo-enolized hydroxy-*o*-quinodimethane species based on our further mechanistic studies. Firstly, we found that 2-methylbenzaldehyde substrates lacking a methoxy group *ortho* to the aldehyde showed no

reactivity in this photoenolization/Diels–Alder reaction, such as the substrates **C1-4** in Figure 1. This identifies the *ortho* methoxy group as crucial for the reaction, consistent with observations by Nicolaou and coworkers. We considered that the photoenolized hydroxy-*o*-quinodimethane and *ortho* methoxy group may be chelated by $\text{Ti}(\text{O}i\text{-Pr})_4$, forming a relatively stable complex. The *ortho* methoxy may serve as a key neighboring group that helps to stabilize the short-lived photoenolized hydroxy-*o*-quinodimethane diene.

Figure 1. Substrates lacking a methoxy group *ortho* to the aldehyde.

- (3) To gain further insights into the reaction mechanism, we carefully investigated the effects of dienophile and $\text{Ti}(\text{O}i\text{-Pr})_4$ dosage on the reaction yield using the model reaction between **7** and **8**. We found that increasing the dosage and concentration of dienophile **8** had little effect on the reaction rate or yield (left curve, Figure 2). In contrast, the reaction yield depended strongly on the dosage of $\text{Ti}(\text{O}i\text{-Pr})_4$. Photolysis using $\text{Ti}(\text{O}i\text{-Pr})_4$ dosages >2.0 equiv. gave stable and comparable yield (right curve, Figure 2), while decreasing the dosage dramatically reduced the reaction yield. Using 50 mol % $\text{Ti}(\text{O}i\text{-Pr})_4$ produced the cycloaddition product **9** in only 5.7% yield. These findings suggest that the reaction intermediates of diene and dienophile may interact with twice amounts of $\text{Ti}(\text{O}i\text{-Pr})_4$ during the photoreaction.

Figure 2. Effects of dienophile and $\text{Ti}(\text{O}i\text{-Pr})_4$ dosage on the reaction yield.

- (4) We also tried to study the process of this PEDA reaction using NMR spectrum (Figure 3). A mixture of **7**, **8** and $\text{Ti}(\text{O}i\text{-Pr})_4$ in toluene- d_8 was irradiated with UV light under the optimized conditions. We monitored the reaction using NMR every 5 minutes without quenching the reaction by saturated sodium bicarbonate. We found that the cycloaddition product **9** formed quickly during photolysis, and that it existed mainly as a

Ti-chelated complex based on comparison with the NMR spectrum of purified **9**. This Ti-chelated complex could be transformed to product **9** by treatment with sat. NaHCO₃. This clearly indicates that Ti(Oi-Pr)₄ plays a key role in this photoreaction, and that a chelation between Ti(Oi-Pr)₄ and diene/dienophile occurs during this process.

Figure 3. Monitor the PEDA reaction with NMR spectrum.

- (5) Given these results and structural data on PEDA products, we propose plausible transition states for the reaction as shown in Scheme 1, which presents the formation of **9** as an example. We consider that the hydroxy-*o*-quinodimethane species is effectively generated via photoenolization, after which the *Z*-dienol and *ortho* methoxy group may be chelated by Ti(Oi-Pr)₄, forming a relatively stable complex. This complex may exist as a monomeric (C5) or dimeric titanium complex (C6). Although we cannot rule out the possibility of a monomeric form, we think the dimeric form C6 is more likely, given the observed relationships between Ti dosage and reaction yield. The *ortho* methoxy may serve as a key neighboring group that helps to stabilize the short-lived photoenolized hydroxy-*o*-quinodimethane diene, which then interacts with a cyclic dienophile such as **8** to give a chelated intermediate C7. Then the activated enone reacts with the diene component from the *endo* direction, forming the Ti-chelated complex of **9**, which can be detected by NMR (Shown in Figure 3). After dissociation, cycloaddition product **9** with three consecutive stereogenic centers is generated stereospecifically.

Scheme 1. Proposed plausible transition states in the PEDA reaction.

(6) We added the mechanistic studies in the revised Supplementary information (S87-88).

4. The “Ti-promoted photoenolization Diels-Alder” (in red in the TOC graphics) could be interpreted in more than one way – the statement needs to be more concise.

“Ti-promoted photoenolization Diels-Alder” was replaced with “Ti(Oi-Pr)₄-promoted Photoenolization/Diels–Alder”

5. On page 3 the authors describe a photochemical experiment in the absence of a Lewis acid: “This suggested that the active hydroxy-*o*-quinodimethane species of **7** was generated and consumed soon after photoenolization.” The word “consumed” is somewhat imprecise. Is there a side product, or the photogenerated quinodimethane simply reverts back to the starting ketone via the ground state proton transfer?

We appreciate this comment. We initially began our study on photoenolized Diels-Alder reaction with combining the model substrate 3,6-dimethoxy-2-methylbenzaldehyde (**7**) with less hindered methyl vinyl ketone under Nicolaou’s photo-reaction conditions, it did really work to give the desired cycloaddition product. This suggested that the active hydroxy-*o*-quinodimethane species of **7** was generated and could react well with less hindered dienophile (methyl vinyl ketone). While conducting the photo reaction with the sterically hindered dienophile **8** (3-methyl-2-cyclohexene-1) only led to trace amounts of hetero-cycloaddition product **11** and no formation of the desired cycloaddition product. So, we thought that the active hydroxy-*o*-quinodimethane species of **7** was generated and decomposed soon after photoenolization, before it could interact with the sterically hindered dienophile **8** based on the above experiment. We replaced “consumed” with “decomposed”.

6. The excessive precision for the wavelength (366nm) of irradiation should be corrected. If the authors used Rayonet RPR-3500 lamps, they emit in a broad range from 300 to 400nm, with the max emission somewhere in the middle. The precise label “366nm” is only applicable to narrow bandpass filters, or monochromators, or +/- 5nm UV-LEDs.

We thank this reviewer for her/his kind reminding. The photo reactor used for this photolysis is Rayonet RPR-200

(Southern New England Ultraviolet Company). As mentioned by this reviewer, the emission spectra of the 16 lamps for 366 nm in the Rayonet chamber reactor do range from 300 to 400 nm with maximum emission wavelength at 366 nm. Therefore, “366 nm” in the manuscript and the Supplementary Methods was all replaced with “ $\lambda_{\text{max}} = 366 \text{ nm}$ ”, detailed instructions were added in General Experimental Procedures of the Supplementary Methods.

7. There was a minor misassignment of carbons in the Phytochemistry 1995 paper on oncocalyxone B: two pairs need a swap: C4a-C9a and C6-C7. C4a should have chemical shift of 137.3ppm and the adjacent C9a – 143.1ppm C6: 15.5ppm; C7: 20.4ppm. To double-check the assignment, the authors could use any method for computing ^{13}C shifts listed on Dean Tantillo’s webpage cheshirenmr.info. Alternatively, a simple HMBC would show that the carbon at 137.3ppm has an HMBC cross peak with H3 (which is only possible for C4a, not C9a).

We really appreciate this reviewer for pointing out the misassignment of the NMR spectrum of oncocalyxone B. As suggested, we run the HMBC spectrum of Oncocalyxone B, which is shown in Figure 4. After carefully analyzing the HMBC spectrum, we conclude that there was only one pair [C4a-C9a] need to swap. We found that the carbon at 137.3 ppm do have an HMBC cross peak with H3, which is only possible for C4a, not C9a. Therefore, C4a should have chemical shift of 137.3 ppm and the adjacent C9a have chemical shift of 143.1ppm. For C6-C7, we only observe that H8 have a faintish correlation peak with the carbon at 15.5 ppm and no cross peak with the carbon at 20.4 ppm. So, we think the carbon at 15.5 ppm is only possible for C7. That is to say, there is no misassignment between C7 and C6. The HMBC spectrum of Oncocalyxone B has also been added in Supplementary (Figure 83).

Figure 4. HMBC spectrum for Oncocalyxone B (6).

8. "protonic" solvent -> protic

It was corrected as suggested.

Response to Reviewer 3's Comments

1. p1. Change sentence "The additional quaternary carbons contribute and demonstrate the diversity of these challenging chemical structures" to "The additional quaternary carbons contribute to the diversity of these challenging chemical structures."

It was corrected as suggested.

2. p.3 and throughout the manuscript, replace "stereo-hindered" with "hindered" or "sterically hindered"

As suggested, "stereo-hindered" was replaced with "sterically hindered" in the manuscript.

3. p3: replace "we realized that the major challenges rely on:" with "we realized that the major challenges were:"

It was corrected as suggested.

4. p3: replace "low-reactive" with unreactive or poorly reactive.

As suggested, "low-reactive" was replaced with "poorly reactive" in the manuscript.

5. p4. "We found that using a catalytic amount (0.2 equiv.) produced only trace amounts of **9**, while using >2.0 equiv. provided optimal yield. These results suggest that the reaction is driven by complexation of Ti(O*i*-Pr)₄ with both reactants **7** and **8**." It is not clear why the authors conclude that complexation with BOTH is occurring, simple based on stoichiometry of the Ti reagent, as there could be many other explanations. The only solid conclusion is that the Ti reagent is required, and there is a potential complexation with at least one cycloaddition counterpart. In the absence of more rigorous physical evidence, the sentence needs to be changed to reflect observations more precisely.

We appreciate this comment. To explore the mechanism of this photo reaction, we further studied: (1) the effects of dienophile and Ti(O*i*-Pr)₄ dosage on the reaction yield, and (2) the process of this PEDDA reaction using NMR spectrum, which have been shown and described above. Based on these studies and this reviewer's suggestion, we removed this sentence and a more detailed description is added in the manuscript (last paragraph, page 6). During our studies of reaction scope, we found that 2-methylbenzaldehyde substrates lacking a methoxy group *ortho* to the aldehyde showed no reactivity in this photoenolization/Diels–Alder reaction. This identifies the *ortho* methoxy group as crucial for the reaction, consistent with observations by Nicolaou and coworkers. The *ortho* methoxy group was not required in the case of less sterically hindered dienophiles, as the Nicolaou group also found. We considered that the photoenolized hydroxy-*o*-quinodimethane and *ortho* methoxy group may be chelated by Ti(O*i*-Pr)₄, forming a relatively stable complex. The *ortho* methoxy may serve as a key neighboring group that helps to stabilize the short-lived photoenolized hydroxy-*o*-quinodimethane diene. Meanwhile, according to the findings of conditions screening, using >2.0 equiv. Ti(O*i*-Pr)₄ provided optimal yield. These results suggest, taking

model substrate as an example, that the reaction is driven by complexation of $\text{Ti}(\text{O}i\text{-Pr})_4$ with both reactants **7** and **8**.

REVIEWERS' COMMENTS:

Reviewer #2 (Remarks to the Author):

I have examined the revised manuscript and feel that the authors have addressed most of the points that I raised. While I agree with the reviewer 1 that the D.-A. reaction of photoenols has been around for long time (btw – it would not be a bad idea to acknowledge the contributions of Pete Wagner's group in this area), the authors have addressed a very important issue of complexity which warrants publication.

I am still not sure if "Ti-promoted photenolization/Diels-Alder" accurately conveys the main point of this manuscript, but this is a minor point